# Companion Cats Show No Effect of Trial-and-Error Learning Compared to Dogs in a Transparent-Obstacle Detour Task

**DOI:** 10.3390/ani13010032

**Published:** 2022-12-21

**Authors:** Muhzina Shajid Pyari, Kata Vékony, Stefania Uccheddu, Péter Pongrácz

**Affiliations:** 1Department of Ethology, ELTE Eötvös Loránd Tudományegyetem—Eötvös Loránd University, Pázmány Péter Sétány 1/c, 1117 Budapest, Hungary; 2Comparative Ethology Research Group, MTA-ELTE Magyar Tudományos Akadémia-Eötvös Loránd Tudományegyetem, Pázmány Péter Sétány 1/c, 1117 Budapest, Hungary

**Keywords:** cat, dog, detour, transparent obstacle, laser pointer, experience

## Abstract

**Simple Summary:**

Although both companion dogs and companion cats are increasingly popular subjects in cognitive behavioral sciences, there is a relative lack of such experiments where their performance is directly compared in the same problem-solving task. In this research, we tested privately owned cats and similar-sized (‘cat-sized’) companion dogs in the well-known detour task. Cats and dogs had to negotiate a V-shaped transparent mesh fence if they wanted to reach the reward placed in the fence’s inner corner. We found that dogs mastered the task faster than cats did, and dogs tended to run along the same side of the fence where they had previous success. Cats, on the other hand, often switched sides. We argue that for dogs, a detour is a harder task than for cats; therefore, it is more important for dogs to learn from their previous successful experiences than it is for cats. Another difference was observed between the two species’ behavior, as the dogs glanced more frequently towards their owner than the cats did during the task. This might indicate that either the task was harder for dogs (thus eliciting more ‘social referencing’) or cats are less dependent on humans than dogs are.

**Abstract:**

We tested companion cats and dogs in similar indoor conditions using identical procedures in the classic detour task around a V-shaped transparent wire-mesh fence. Besides the control group, we used two types of laser light-pointing demonstration (moving around the fence, or pointing straight at the reward). We found that dogs reached the food reward faster than cats; across consecutive trials, only the dogs showed improvement in their speed and dogs continued to use the same side for detouring after a preceding successful attempt, while cats chose the side for detouring irrespective of their previous successful trials. In addition, ‘demonstrating’ a detour with the laser did not influence the speed or direction of the detour of the subjects; and dogs looked back to their owner more frequently than the cats did. We discuss the possibility that for dogs, detouring along a transparent obstacle represents a more problematic task than for cats; therefore, dogs strongly rely on their previous experiences. This is the first time that cats were successfully tested in this detour paradigm in direct comparison with dogs. The results are relevant from the aspect of testing cognitive performance in companion cats, which are known to be notoriously reluctant to engage with novel experimental situations.

## 1. Introduction

The so-called detour paradigm is an ecologically valid and versatile tool used to study diverse cognitive skills in various species (for a review, see [1]). Many species face more or less often the challenge of detouring around obstacles in their natural environments. The difficulty of a detour is due to the natural inclination to choose the shortest (straight) route to reach a distant visible goal. This so-called ‘perceptual magnet’ effect is easily noticed when opaque barriers represent an easier-to-detour obstacle for the same species than a transparent barrier (e.g., in dogs [2]; or cats [3]). Similarly, distance to the goal affects detour behavior. A closer goal makes it harder to move away from it temporarily while the animal performs the detour, but increased goal distance makes it easier to execute a detour (chickens: [4]; dogs: [5]). Motivation [4,6] to solve the task, age of the animal [7,8], sensory capacity (various species of fish [9]), raising conditions [10] and orientation/layout of the barrier [11,12,13] are among the numerous factors that can also affect the detour performance. 

Being able to choose between different paths to reach a distant goal (example: reaching the nest, escaping from a predator, capturing fleeing prey, etc.) is part of the natural skillset of most species under natural conditions. There can be various pathways to reach the goal, but it may provide survival value for an animal to choose the most appropriate route. Perception of the task features might differ due to ecological constraints in a given species; in other words, the ecology of a species can influence its typical behavior in a detour situation. For example, dogs perform better in the case of an outward detour than during an inward detour around a V-shaped barrier, most likely because they tend to avoid confined spaces [12]. 

A detour problem can be more ecologically relevant for some predators, if their pursuit of prey often involves obstacles and multiple routes to reach the goal [14]. For example, jumping spiders navigate their complex three-dimensional environment to pursue prey using their well-developed visual system [15]. Cross and Jackson [16] have tested jumping spiders in a detour task, where the spiders from the starting position could observe a target placed on top of a pole. Although the subjects were only allowed to take a roundabout walking route without the goal in sight, instead of a direct jump to reach the target, they still had a high success rate. Many species have been studied in various detour experiments, where their success rates showed an association with the assumed evolutionary need to be adapted, for solving this type of problem ([17]; canids: [12,18,19]; great apes: [20]; monkeys: [21]; reptiles: [22]).

Learning effects and individual variations within species can also influence detour success [5,23,24,25]. Dogs only moderately improved their detour performance via a trial-and-error type of learning around a V-shaped transparent fence (e.g., [12,26,27]). At the same time, dogs can readily learn how to detour by observing either human or conspecific demonstrators [28,29]. However, it is interesting to note that dogs copy the route of the demonstrator only if they had no experience with successful detours before, as they tend to stick to the directional component (i.e., detour along the left or right side of the obstacle) of their own previous detours [26]. The reluctance to abandon a previously successful detour method by dogs was also confirmed with other types (straight barrier) of detour devices [30]. There are indications that the success in a detour task may be connected to the subjects’ inhibitory control capacity. Bray and colleagues [31] found that assistance dogs (bred/trained for excellent inhibitory control) perform better than average pet dogs when the detour task involves high arousal-inducing motivation. 

Domestic cats are skilled predators, in which access to the outdoors, early maternal support and experience with real prey can positively affect hunting skills [32]. Cats with established companion status and regular food provisions still habitually hunt and kill prey animals [33,34]. Several studies have shown that in a visible displacement test of object permanence, cats [35,36,37,38] show high success rates, but they were unable to find an object when they were tested for invisible displacement tests [35,36,39,40,41]. Dumas [42] used a methodology that was biologically more relevant to cats to study invisible displacement tasks and found that the cats demonstrated a high success rate in solving the problem. In this particular test, cats watched a target object from behind a transparent panel, but they had to take a detour around an opaque panel to actually reach the target. While the cats were behind the opaque panel, unknown to them, the object was (invisibly) hidden behind one of two screens. The performance of the tested subjects suggests that cats possess cognitive capacities to deal with complex events that may occur while they pursue prey; in other words, cats can successfully anticipate the invisible displacement of a target during pursuit. In addition, it is known that the reverse learning phase of the T-maze demonstrates cognitive flexibility in cats [43].

Recently, domestic dogs were tested several times with the detour paradigm, where researchers capitalized on the gregariousness of this species and on the complex dog–human relationship through various social learning tasks [44]. Conversely, our knowledge about cats’ behavior in the detour task is very limited, apart from a few early publications, which provided information about the role of transparent and opaque obstacles in cats’ success [3]. However, to our best knowledge, cats and dogs so far have not been compared under similar detour conditions. In this study, we included companion cats and dogs in a comparative experimental design, where we tested both cats and dogs with the same method, under identical detour conditions using a V-shaped transparent wire-mesh fence. This fence was chosen because its versatility has been copiously proven in earlier papers published on individual and social learning of the detour task by dogs [26,27,28,29]. Additionally, we wanted to provide a rather difficult task to the subjects; therefore, the transparent fence seemed to be a superior option to the opaque versions [2,3]. We opted for asocial test stimuli (i.e., no social learning tests were performed at this time) because we wanted to concentrate on ecologically relevant factors in both species. Although dogs are well known for their ubiquitous willingness to interact and rely on human demonstrators in social learning tasks [44], in the case of cats, there is much less unequivocal evidence available regarding their dependence on human behavior [45]. Furthermore, as the size ratio between the obstacle and the animal seems to be an important factor when determining the difficulty of the task, we recruited only small-sized (i.e., ‘cat-sized’) dogs in this experiment. The size of the participating cats was rather uniform, which was expected from a sample that consisted mostly of generic European shorthair mongrel cats (for further details on small variations in mixed-breed cats’ size, see [46]). Our main goals were (a) to investigate whether the transparent V-shaped fence is a suitable method for testing cats with the detour paradigm, (b) whether prey-like asocial stimuli (i.e., stimuli that may elicit attempts to follow, chase or catch) affect cats and dogs in their detour efficiency and detour path and (c) whether there is a difference between cats and dogs in their solutions to these tasks, based on their typical predatory behaviors. We predicted that cats and dogs will show similar detour efficiencies to each other on their own, because in earlier studies, dogs were found to be rather inefficient at this type of task without the option for social learning [12], and cats performed with low success rates against transparent detour obstacles [3]. For prey-like stimulus, we opted for the light of a laser pointer. At least in the case of cats, there is evidence that this type of stimulus elicits the typical locomotor components of the hunting sequence (stare, chase and pounce) [47], and dogs have also been shown to be interested in visual stimulation by static or moving laser dots [48]. However, when the subjects could observe the red dot from a laser pointer that moved in a ‘detouring’ fashion around the V-shaped fence, we predicted that cats would perform better than dogs because this type of stimulus may be more suitable for cats more than for dogs, as dogs are considered to be more motivated by social stimuli [28]. We predicted that the directional element of detouring (left vs. right side of the fence) would be influenced more by their own experience in the case of dogs, and cats would be more likely to rely on the route taken by the prey-like stimulus. Finally, we predicted that the dogs would rely more on the (passive) presence of their owner during problem-solving; thus, we expected the dogs to look back to their owner more frequently than cats would.

## 2. Materials and Methods

### 2.1. Subjects

#### 2.1.1. Companion Cats

Participants were recruited through advertisements on the website of our department and via social media. Criteria for inclusion were simple and any companion cat could participate in our tests if the cat was at least 6 months old, and was not physically challenged (for example, impaired vision and hearing, permanently injured, etc., which were all based on the owners’ report about the health status of their cats). After exclusions, the effective sample size was 53 subjects (6 intact cats and 47 neutered/spayed cats), of which all were tested and included in the analysis (mean age: 53.74 months, minimum age: 6 months, maximum age: 210 months; 34.50 ± 3.1 cm height). Most of our subjects were European Shorthair cats (N = 45), plus two Persians, two Siamese, two Oriental Shorthairs, one Birman and one Abyssinian cat. 

#### 2.1.2. Companion Dogs

Participants were recruited through advertisements on the website of our department and via social media. As we wanted to test the subjects from both species with exactly the same detour obstacle, we planned to keep the size ratio between the subjects and the detour device the same. Thus, only small- or small–medium-sized dogs with a maximum height of 45 cm (height at the top of the head in a normal standing position) participated in our tests. This size range fits the average size of a typical adult mixed-breed cat (the majority of our cat sample) well. We included dogs independently of their purebred status (28 purebred dogs and 10 mixed-breed/mongrels). All the subjects were 8 months old or above, as dogs’ performance in the detour tests seems to be rather robust and independent of age above 6 months [49,50]. After exclusions, the effective sample size was 38 subjects (2 intact dogs and 36 neutered/spayed dogs) and these were all tested and included in the analysis (mean age: 61.36 months, minimum age: 8 months, maximum age: 168 months; 29.56 ± 6.2 cm height). 

### 2.2. Exclusions

#### 2.2.1. Companion Cats

Originally, we tested 58 companion cats. Subsequently, five cats were excluded from the final analysis because of too-loud background noise (3 subjects) (example, construction noises from neighboring houses; noise from other pets at home) at the testing location, or because the subject hid during the test (2 subjects).

#### 2.2.2. Companion Dogs

Originally, we tested 42 companion dogs. From these, four dogs were excluded from the final analysis because these dogs did not approach the experimental apparatus.

### 2.3. Procedure

Each test was performed indoors. Cats were tested in the home of their owners, because in the case of most companion cats, testing them away from home (e.g., in an unknown testing room at the institute) generates strong stress that would impede their successful participation in the tests (for other instances of cats being tested at their owners’ homes, see [51,52,53]). Each test was recorded with a Panasonic HDC-SD10 (Panasonic Corporation, Kadoma, Osaka, Japan) and Moto G (Motorola Mobility LLC, Chicago, IL, USA) mobile phone camera (Panasonic was replaced with Moto G in the later stage of data collection as it was damaged) camcorder for later analysis. The experiment consisted of three trials per subject. For controlling the hunger level of the participating animals, we requested that the owners withheld food from the subjects for 5 hours prior to the test.

Companion dogs were tested in the laboratory of the Department of Ethology at Eötvös Loránd University, as it has been well established that these animals can be reliably tested (especially in the presence of their owners) away from home (e.g., in the laboratories of universities, or at outside locations, such as dog schools), with the universally recognized standard protocol outlined in this study (e.g., [54,55,56]). Furthermore, testing the dogs in the laboratory (away from their homes) made these results readily comparable with other tests on dogs. Finally, some dogs can be territorial at their homes with visiting strangers, which could potentially cause problems when the experimenter arrives to test them. Therefore, we did not need to test the dogs in their owners’ homes to obtain comparable results to cats. Each test was recorded with a Panasonic HDC-SD10 camcorder for later analysis. 

A transparent V-shaped fence made of plastic mesh on a lightweight steel frame, with dimensions of 1m height, 1.5 m length and 90-degree angle at the intersection, was used as an obstacle to provide the need for detouring to reach the target food. Companion cats and companion dogs were all tested with the same procedure. 

There were three test groups, which were as follows: the control group, straight-pointing demonstration group and detour-pointing demonstration group. A laser pointer was used as prey-like stimulus in the detour-pointing demonstration and straight-pointing demonstration groups. In the case of the two test groups with the laser stimulus, before the test trials, we provided a short warm-up for each subject with the laser pointer. The main reason for this was to make sure that the subjects were interested in the red laser dot projected on the floor. While the subject was gently held by the owner, the experimenter directed the red laser dot on the floor approximately 0.3 m from the subject, and then moved it slowly away from the subject by another 0.3 m. The subject was considered as ‘interested’ in the laser stimulation if it stared at the dot and kept watching it during its movement. All the cats and dogs passed this stage. Each subject was randomly assigned to one of the test groups. Each subject was tested only once and could belong to only one test group.

Tests consisted of three consecutive 1 min trials in each group. There is ample evidence in the case of dogs that three trials are enough to detect performance changes (improvement), if the subjects are provided the chance to learn via observation [12,26,27,28,29,49,57]. Because the performance of cats was previously unknown in the V-shape fence detouring paradigm, we also opted for 3-trial test sessions for the cats in order to meet the goal of drawing direct parallels between the two species. Additionally, as cats are prone to losing interest in repetitive trials during experiments (e.g., [51]), the possible shortest test setup also had additional advantages from this aspect.

#### 2.3.1. Control Group

For trials 1, 2 and 3, the experimenter asked the owner to stay with the subject at the start point (at the mid-line, 1 meter away from the intersection of the V-shaped fence (see Figure 1a,b)). The owners were asked to keep the subjects at the start point by gently restraining them. The experimenter showed a bowl with food to the subject and let the subject sniff the food. We offered the option to the owner to provide their preferred food for their cats and dogs. If the owner did not have a preferred food, we used Purina Pro Plan Nature Elements Spirulina for dogs and Purina Pro Plan Complete Essentials (Nestlé Purina Petcare, St. Louis, MO, USA) for cats. The experimenter placed the food in the inside corner of the fence. Then, the experimenter moved back to the start line next to the owner and asked the owner to release the subject. The subject was free to make a detour to reach the food target for a maximum duration of 1 minute. If the subject did not succeed in reaching the food within 1 minute, the owner took the subject back to the start point. If the subject made the detour and reached the food within 1 minute, the subject was allowed to eat the food, before continuing to the next trial. The owners had to remain at the start point during the trials, and they were not allowed to direct the subject with hand signals. However, verbal encouragement was allowed. 

#### 2.3.2. Straight-Pointing Group

Trial 1 was identical to Trial 1 in the control group.

Trial 2 ran identically to Trial 1 to the point the food was placed behind the fence.

The experimenter then returned to the start point, and switched on the laser pointer, positioning the red dot on the floor directly in front of the subject and then moved the red dot towards the food target in a straight line. When the red dot reached the target, it remained pointing at the target for 3 s. The owner had to gently hold the subject at the start point, and she/he was asked to release the subject as soon as the experimenter turned off the laser pointer. From this point forward, the trial ran identically to Trial 1. 

Trial 3 was identical to Trial 2.

#### 2.3.3. Detour-Pointing Group

Trial 1 was identical to the trials in the control group and Trial 1 of the straight-pointing demonstration group.

Trial 2 ran identically to Trial 1, to the point the food was placed behind the fence. The experimenter returned to the start point, turned on the laser pointer and positioned the red dot on the floor directly in front of the subject. After this, the experimenter slowly moved the red dot to the fence in a straight line, then along one side of the fence in a ‘detouring fashion’ by at first moving to the outer side end, then turning around and then moving along the inner side to the front corner where the food was located. Finally, the red dot paused on the food target for 3 seconds. It was important for the experimenter to always move the laser pointer along the opposite side of the fence than the one chosen by the subject in Trial 1 (e.g., if the subject performed the detour on the right side of the fence to reach the target in Trial 1, the laser-pointer demonstration was carried out on the left side of the fence to the target). If the subject did not perform a successful detour in Trial 1, the experimenter randomly chose the left or right side of the fence for the laser demonstration. The owner had to gently hold the subject at the start point, and she/he was asked to release the subject as soon as the experimenter turned off the laser pointer. From this point on, the trial ran identically to Trial 1.

Trial 3 was identical to Trial 2, including the side choice for laser demonstration (meaning that the experimenter chose the same side as in Trial 2 for the laser demonstration).

### 2.4. Data Collection

The basic demographic details of the dogs and cats were collected from the consent form filled out by the owner before the tests. The behavior of the subjects was coded from the video segments with the help of Solomon Coder (beta 17.03.22 copyright by András Péter). Table 1 shows the list of coded behaviors and their description. Twenty-five percent of the total recorded videos of those included in the final analysis were coded by a second observer and interrater reliability scores were calculated in SPSS 28 (Statistical Package for Social Sciences–IBM, Armonk, NY, USA) for all the parameters included in the results. Kappa coefficients were estimated to be in the range between 0.7 and 0.8 for all the parameters included in the results. 

### 2.5. Statistical Analysis

We compared the success rate of the subjects for Trial 1 across all three experimental groups using Kruskal–Wallis tests with post-hoc tests in SPSS 28. In addition, we compared the summarized success rates of Trials 2 and 3 for all the subjects across all three experimental groups.

We used R statistical software (version 4.0.5, R Development Core Team, 2015) in RStudio (RStudio Team, Boston, MA, USA) for the analysis of latencies.

The detour latencies were analyzed using mixed effects Cox regression (coxme function) [58]. (We checked the main effects and all the possible two-way interactions of the variables. We ran the model for latency to reach food in which the species type, demonstration type, and number of trials were fixed effects, and the ID of the subjects was a random effect. To find the best fitted model, we used bottom-up model selection using ANOVA and we always used the results of the most parsimonious (final) models. For the pairwise comparisons, we ran Tukey post-hoc tests (emmeans package), where we compared the type of species and number of trials.

We also noted the direction of the dogs’ and cats’ detours. We analyzed the concordance of both species’ detours under all three test conditions in relation to the direction of Trial 1 with Wilcoxon one-sample tests in SPSS 28. Subjects with an unsuccessful Trial 1 were excluded from this analysis. In the control and straight demonstration groups, we hypothesized that in the case of cats, there would be no significant concordance between the directions of the detours across the trials (i.e., cats choose their detour directions randomly). In the case of dogs, based on the findings of Pongrácz et al. [12], we hypothesized that the dogs would follow the same direction in Trials 2 and 3 as in Trial 1. Under detour laser-pointer demonstration, we hypothesized that cats would choose the same side as the laser demonstration side, but dogs would not.

Looking back to the owner was only sporadically observed, especially in cats; therefore, we report these results only as descriptions.

## 3. Results

We did not find any significant differences between the success rate in Trial 1 across any of the three experimental groups for cats or dogs (cats; H(2) = 4.58; *p* = 0.1, dogs; H(2) = 0.647; *p* = 0.724). Similarly, the combined success rates of Trial 2 and 3 for both species were not significantly different (cats: H(2) = 3.902; *p* = 0.142, dogs; H(2) = 0.881; *p* = 0.644).

Independently of the test group, dogs solved the task significantly faster than cats (exp(ß) = 5.86; 95% CI = (2.568–13.383); z = 2.32; *p* < 0.0001) (Figure 2). We also found an interaction between the species and trial number, as the post-hoc tests showed that dogs were significantly faster in the third trial than in the first trial (exp(ß) = 0.363; 95% CI (0.200–0.658); z = −3.342; *p* = 0.0024) (Figure 3), while the latency of reaching the food did not change significantly in the case of cats (*p* = 0.4182) (Figure 4). The interrater reliability score (Kappa) for the latency of detours was 0.79 and 0.72 for cats and dogs, respectively.

Concordance in the direction of detours was analyzed only for cats and dogs that reached the target within the time limit in the first trial (successful Trial 1). We used the number of subsequent trials in which concordance occurred in relation to the direction of the first trial. A Wilcoxon one-sample test revealed that the cats did not walk on the same side of the fence in the subsequent trials, irrespective of the type of demonstration (concordance by chance would be 1; control group: T = 22.5, N = 12; *p* = 0.480 *(*Figure 5*);* straight-pointing group: T = 7.00, N = 8; *p* = 0.414; detour-pointing group: T = 69, N = 17; *p* = 0.718; in this case, concordance by chance would be 0.83, i.e., based on the proportion of subjects in the control group that chose the same direction as in Trial 1, during Trial 2 and Trial 3) *(*Figure 6). In contrast, dogs were loyal to the direction of detour chosen in the first successful Trial, irrespective of the type of demonstration (the hypothetical concordance was 2; control group: T = 0.00, N = 12; *p* = 0.063 *(*Figure 5*);* straight-pointing group: T = 0.00, N = 10; *p* = 0.317; detour-pointing group: T = 25, N = 9; *p* = 0.763; in the last case, concordance by chance would be 1.3, i.e., the proportion of subjects that chose the same direction as in Trial 1, Trial 2 and Trial 3 in the control group) *(*Figure 6). We also checked whether the subjects followed the direction of the moving laser dot in Trials 2 and 3 in the detour-pointing groups. The hypothetical value was 1. We found no significant concordance between the side where the laser dot was moving and the side choice of cats in Trials 2 and 3 (T = 45.50, N = 18; *p* = 0.564). In the case of dogs, we found no significant concordance between the side where the laser dot was moving and the side chosen for detouring around the fence in Trials 2 and 3 with a hypothetical value of 1 (T = 8.00, N = 9; *p* = 0.257). The interrater reliability score (Kappa) for concordance with T1 and laser demonstration side is 1.00 for both cats and dogs.

Out of 53 cats, only 5 animals looked back at the owner during problem-solving (3 from the control group, 1 from the straight-pointing group; 1 from the detour-pointing group). Out of the five cats, only two animals looked back more than once at the owner (2 and 4 times, respectively). Out of the 38 dog participants, 13 animals looked back at the owner, with 4 dogs looking back more than once (control group: 5, straight-pointing group: 5; detour-pointing group: 3).

## 4. Discussion

We tested companion cats and dogs in similar indoor conditions, using identical procedures in the classic detour task around a V-shaped transparent wire-mesh fence (e.g., [12,26,27]). To our knowledge, this is the first time that cats have been successfully tested in this detour paradigm in direct comparison with dogs. Our experiments showed that (1) dogs were faster to reach the food reward than cats; (2) the dogs, but not the cats, showed improvement in their detour latencies across the consecutive trials; (3) in the case of a successful first trial, the dogs continued to use the same side for detouring in the subsequent trials, (4) while cats chose the side for detouring irrespective of their first successful trials; and (5) moving a laser pointer around the fence, thus ‘demonstrating’ a potential detour path, neither improved the speed of the detour, nor influenced the side choice of dogs or cats. Finally, 33% of the dog participants showed referential looking back to their owner behavior during problem-solving, while the proportion of cats looking back at their owner remained below 10%.

The involvement of companion cats in cognitive tests often suffers from the high dropout rate of subjects due to the difficulty of finding effective motivation for cats [59]; and their refusal to cooperate with the experimenter [51]. Recently, testing cats in their owners’ home proved to be a promising approach for comparative ethologists (e.g., [52,53]); thus, the satisfactory level of motivation and confidence of our feline subjects in the detour tests could partially be explained by the choice of home-based testing. The fact that companion dogs were tested at the university’s laboratory does not make the testing conditions markedly different because the most important aspect of the ‘natural environment’ for (socialized) dogs is the presence of their owner. Therefore, their behavior and cognitive skills can be reliably tested in any suitable environment if their owner is present [60]. On the other hand, for the sake of comparable experimental conditions, we also chose indoor testing for the dogs. However, indoor testing required the shortening of the original 3 m long outdoor fence that was used in several earlier studies (e.g., [12,18]. To keep the dimension ratio similar for cats, we tested small (i.e., ‘cat-sized’) dog breeds only; however, we found that in our current study, dogs encountered less difficulty when solving the detour task on their own within 1 min than they did with the original outdoor equipment [12]. This is clearly noticeable as the dogs showed no improvement in the detour latency when they had to detour the 3 m long fence three times, without demonstration (control conditions, [12,28,49]), but in this study, they showed significantly shorter latencies in Trial 3, independent from the test group. One possible explanation is that dogs can more easily recognize the distal end of a 1.5 m long fence than a 3 m long fence. For this reason, the maximum trial length could be reduced to 30 s instead of 1 min, when dogs are tested with a shorter indoor fence in future studies. On the other hand, it is also interesting to note that in the current experiment, we only tested small (‘cat-sized’) dogs, meanwhile, in each of the earlier studies, testing was performed independently of subject size. This fact highlights that in dogs, a V-shaped obstacle with 3 m long sides represents a suitably difficult task for every dog; however, even for small dogs, a 1.5 m long V-shaped fence is an easy problem to solve.

For solving a problem such as a detour task, the subject has to be capable of mental planning a detour path that involves a temporary separation from the target. This capacity seems to be independent of the complexity of the central nervous system of the animal, as ‘planned detours’ based on visual observation, and not on repeated trial-and-error learning, were successfully demonstrated in various spiders [16]. It is likely that the effective mastering of a spatial detour task is more closely connected to ecological pressures than to the general cognitive complexity of a species. For example, some authors argue that domestication and subsequent selection of dogs for human dependency may have contributed to the weaker detour performance in dogs compared to their wild relatives (dingoes, [19]; wolves, [18]). From the aspect of comparative cognition, inhibitory control is another frequently targeted capacity with the detour paradigm. Although the version of the test may vary from the straight reach-across type [20] to the spatial-locomotor variant (e.g., [11,18,25], the principle is the same; when the target (reward) is visible across a barrier, the more obvious solution for the subject would be to approach it directly; however, as it is not possible, the subject has to be able to inhibit this reaction and opt for an indirect detour. The fact that cats were, in general, successful in our detour test shows that they can effectively inhibit futile attempts of direct access to the desired reward, which in turn can be well explained by their habit of being ambush predators [61]. For a successful ambush, the hunter has to show patience that requires considerable inhibition against starting the attack prematurely. 

Regarding dogs, our present findings confirmed many of the original findings of a study by Pongrácz et al. [12], in which the dogs managed to perform a detour in the very first trial, after which the dogs remained faithful to the direction of their first successful trial. In addition, across the repetitions, trial-and-error learning improved the performance of the dogs significantly. There is an interesting parallel between these findings and the notion that dogs tend to follow the win–stay/lose–shift strategy in search tasks [62]. From the point of view of ecology, as they are more of a scavenger than a predator, it may be advantageous for dogs to stay with a well-yielding solution over an untried one. However, remarkably, cats did not rely on the direction of their earlier (successful) choice to reach the target. While a detour is considered to be a difficult problem for dogs [12,18,19], where eventual success can lead to a reinforced stereotype in route selection, for cats, detouring around an obstacle might be easier, so they can choose sides randomly. As a predator, they might have realized that both directions around the fence equally lead to the same target. Poucet et al. [3] showed that in a detour problem, cats chose the most optimal path, considering both the distance and angular deviation, to reach the target. In the case of a transparent obstacle, they chose the less divergent path, irrespective of its length. In our case, the V-shaped transparent fence had equal arm lengths and a constant angular deviation for either of the side choices. So, cats might have perceived both of these side choices as equally optimal to reach the target, resulting in no influence of the previous successful choices on the latter side selection or detouring speed. Additionally, cats are typically ambush predators, hunting unpredictable and versatile prey; therefore, for cats, staying at an already exploited spot would be a maladaptive strategy. As a limitation to the aforementioned explanation, we should also remember that the success rates of dogs and cats did not differ; thus, based on this parameter, we cannot state whether the task was harder for the dogs than it was for the cats, as the latter explanation is based only on the side choice patterns of the two species. There are also alternative explanations, emphasizing that dogs might show a more effective trial-and-error learning performance than cats do, because dogs can actually learn to follow the previously successful path behind the fence. This may help dogs to perform the detours more quickly.

Solving of the detour problem was previously combined with the potential utilization of observational learning by the subjects. Dogs were shown to be very effective social learners, both from human and conspecific demonstrators (e.g., [28,29]). The capacity of learning to perform an effective detour from a human was also shown in goats [11]; thus, one can argue that gregarious domesticated animals are able to learn some aspects of their behavior via observation. In our study, instead of providing them with a social learning task, both cats and dogs were tested under the same treatment conditions with laser light demonstration. Although there are recent indications that cats engage in complex socio-cognitive interactions with humans (e.g., attachment-like bonds, [63]; gaze-following, [51]; responding to their name, [52]), at this time, we wanted to compare cats’ and dogs’ responses to non-social stimuli connected to the detour task, because there are still some arguments that there could be fundamental differences between the social bonds formed by the two species with humans [45]. This aspect was also indirectly supported in this study by the descriptive results of the animals looking back at their owner, as more dogs referred back to their owner, and with higher frequency than cats did. As looking at the owner during problem-solving can be interpreted as a sign of social dependency [57,64], the more frequent occurrence of this in dogs can be explained both by evolutionary (i.e., phylogeny and domestication of dogs vs. cats [45]) and environmental factors (i.e., experience with human-assisted problem-solving, as may happen with dogs and socialized wolves [65]). One may assume that the higher frequency in dogs’ looking back behavior was due to the potentially higher responsiveness of dogs to human verbal encouragement. The use of verbal encouragement in detouring and other problem-solving tasks was proven to be of instrumental importance to the success of dogs [44]. It was also shown on several occasions that cats are responsive to verbal utterances of humans [51,52,53]. As it was found in the case of dogs that during the detour task, their looking back frequency depends on the difficulty of the task and not on the verbal utterances of the owner [26,27], it seems unlikely that in the present study, the higher frequency of dogs’ looking back behavior was caused by their stronger reactions to the verbal encouragement from the owners.

The use of laser light pointers is very popular among cat owners for playing and stimulating species-typical predatory behavior. Cats enjoy playing with laser light, as it mimics features of prey in motion (e.g., changes in direction to react to cats’ movement, etc.) [62]. Dogs are also reported to respond to laser-pointer stimulation [66]. It is important to mention that we used laser light for a very short interval with our subjects in order to avoid any negative effect on them. Additionally, during our trials with laser demonstration, laser pointing always ended at the food target, so that the animal would not become frustrated or stressed after searching for the disappearing laser light [67].

Laser light pointing did not have any effect on either species regarding the improvement of their detour performance or influence on their side choice. This result is different compared to our predictions in the case of the cats, as we expected that cats would readily follow the movement (‘route’) of the highly stimulating laser pointer. In the case of the dogs, we expected that the laser pointer would have a weaker effect compared to the social nature of human or conspecific demonstrators [29]. In the case of the cats, the lack of effect was not due to the cats’ inattentiveness, as we made sure to catch the cats’ attention with the laser pointer before the demonstration started. It is more likely that the cats did not follow the laser pointer’s route because (i) they were restrained by the owner until the laser ‘demonstration’ was finished, and they were only interested in the moving light dot until it was clearly visible (therefore, they can ‘chase’ it); or because (ii) detouring for food around the V-shaped obstacle is not a difficult task for cats; therefore, they were not inclined to pay too much attention to other marginal factors that appeared alongside the main obstacle.

The main limitation of this study was the indoor testing condition that forced us to use a shorter than usual fence, thus likely providing the subjects (especially the dogs) with a task that was too easy to solve. This could have caused the laser stimulation to be ignored as the subjects in the control condition, which did not have the laser demonstration, also performed the task successfully. Another potential influencing factor was that the dogs were tested away from home (in the laboratory), while cats were tested at home. However, as dogs still performed rather well in these conditions, we do not believe that the different locations would significantly affect the comparability of the cat and dog tests. Furthermore, in this study, we did not test for the potential effect of observing a social partner (a human or conspecific demonstrator) on the detour performance of cats. This aspect deserves further attention in the future.

## 5. Conclusions

The main conclusion of our study is that an indoor, transparent V-shaped detour fence setup is suitable for testing companion cats on site, providing a promising experimental paradigm that would be useful if developed further, for example, to test the potential capacity of cats in a social learning scenario. By using an identically tested group of companion dogs, we could show that compared to cats, dogs showed a tendency to form a fixed route choice based on their experience of earlier success. The difference between cats and dogs from this aspect may be rooted in the ‘optimizing’ behavior of cats, meaning that if they recognize that both sides of the obstacle represent an equally solvable task, cats will change freely their spatial approach as a result.

## Figures and Tables

**Figure 1 animals-13-00032-f001:**
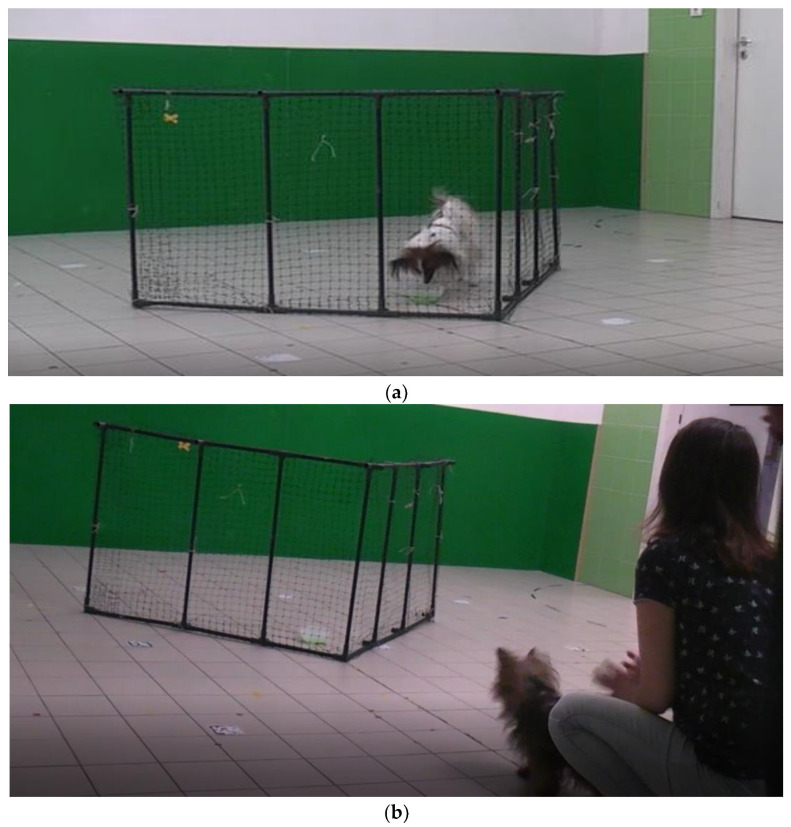
Transparent V-shaped fence (height = 0.5 m, length = 1.5 m, 90-degree angle at the intersection. A dog arrives at the target behind the fence (**a**). A dog is released by its owner. The relative position of the starting point to the fence is visible (**b**).

**Figure 2 animals-13-00032-f002:**
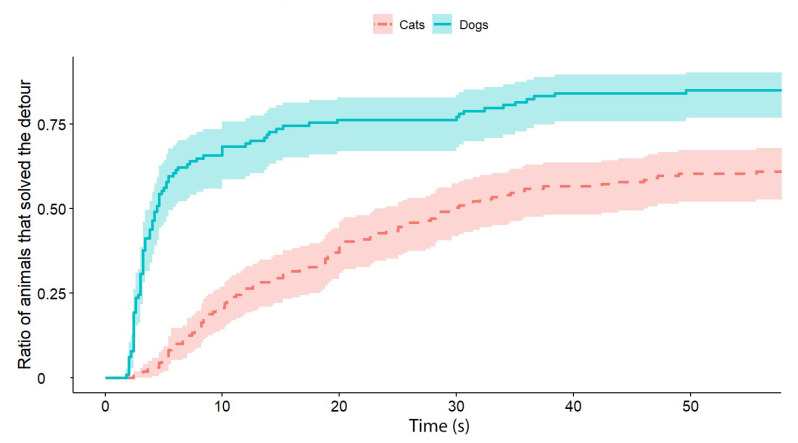
The latency of detours by the cats and dogs (each experimental group, with all Trials together). The latencies of successful detours are indicated on the horizontal axis, and the vertical axis shows the proportion of subjects (cats and dogs separately) that successfully solved the task with a given latency.

**Figure 3 animals-13-00032-f003:**
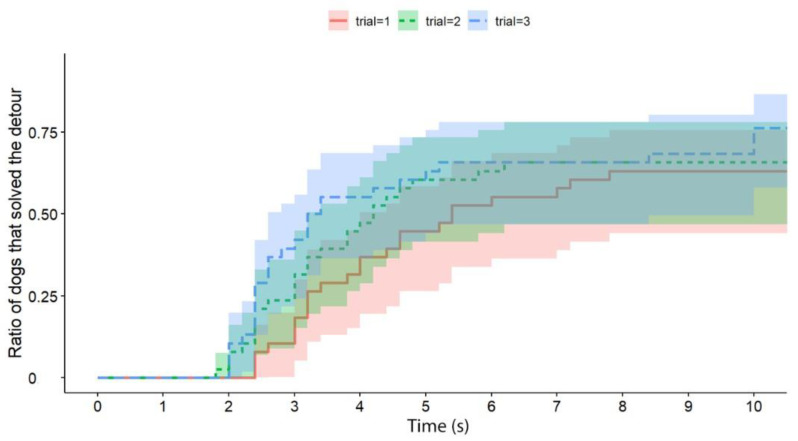
The latency of successful detours by the dogs (all experimental groups together; the Trials are shown separately). The latencies of successful detours are indicated on the horizontal axis, and the vertical axis shows the proportion of subjects (only dogs; the trials are shown separately) that successfully solved the task with a given latency.

**Figure 4 animals-13-00032-f004:**
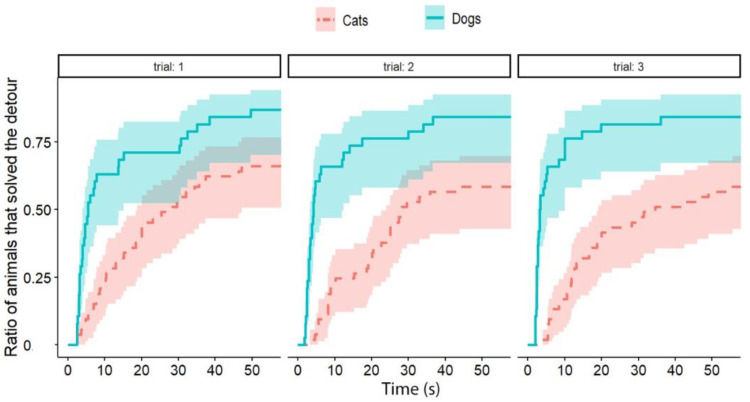
The latency of detours in each Trial by cats and dogs (each experimental group together). The latencies of successful detours are indicated on the horizontal axis, and the vertical axis shows the proportion of subjects (cats and dogs separately) that successfully solved the task with a given latency.

**Figure 5 animals-13-00032-f005:**
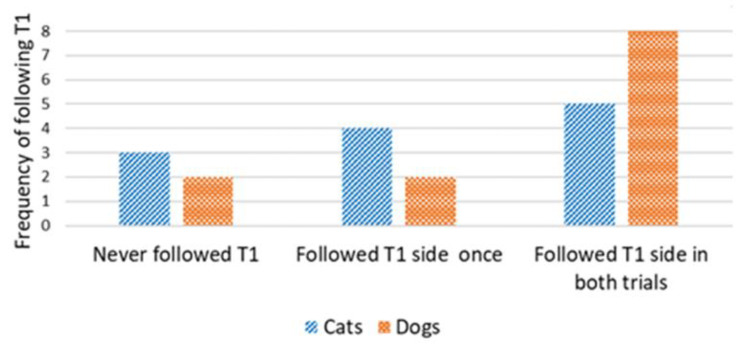
Control group: frequency of following the same side as in Trial 1, during Trial 2 and Trial 3 (cats vs. dogs).

**Figure 6 animals-13-00032-f006:**
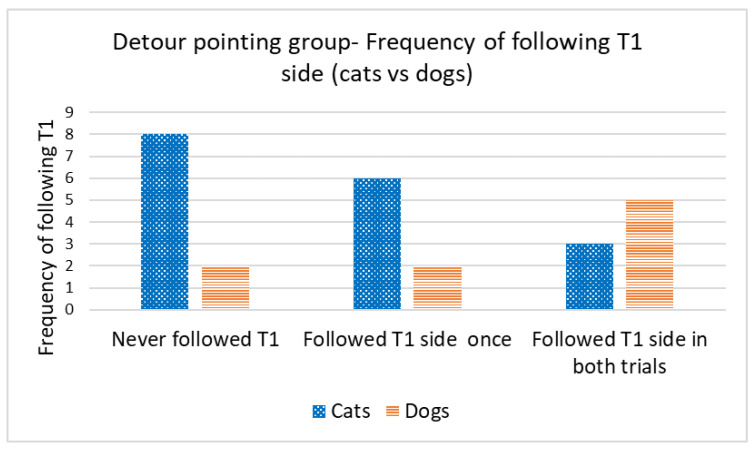
Detour-pointing group: frequency of following the same side as in Trial 1, during Trial 2 and Trial 3 (cats vs. dogs).

**Table 1 animals-13-00032-t001:** List of coded behaviors and descriptions.

Behavior	Description
Reach food/‘success’ (frequency)	After performing the detour, the subject comes in contact with the food (eats/sniffs it).
Latency of detour	The time elapsed from subject release (by the owner) until the subject reached the food. In the case of unsuccessful trials, the maximum value of 60 s is recorded as latency.
Direction of detour (frequency)	The side (right/left) of the V-shaped fence chosen by the subject, eventually leading to reward success.
Looking back (frequency)	The subject turns back (entirely or just with their head) towards the owner, while attempting to detour the fence. This behavior was coded from the release of the subject only until they reached the far end of the fence and turned towards the reward, as from this point on, the subject was facing towards the owner.

## Data Availability

The datasets used and analyzed in the current study are available from the corresponding author upon reasonable request.

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
