# Peer review of "Companion Cats Show No Effect of Trial-and-Error Learning Compared to Dogs in a Transparent-Obstacle Detour Task"

_animals, 2022, doi:10.3390/ani13010032_

Round 1

Reviewer 1 Report

It is great that the authors compared the performance of cats to dogs directly in the same task. I like that they noted whether both species looked to the human and considered multiple interpretations of this finding.

This is a nice sample size for both cats and dogs.

The demonstration manipulation should be justified and described in the introduction. The use of a laser pointer instead of a conspecific demonstration should be more clearly justified in the introduction as well. The authors should address how their procedure deviates from more traditional versions of the detour task using both transparent and opaque barriers.

The authors should discuss whether and why cats and dogs may differ in terms of win stay win shift foraging strategies. Perhaps this makes sense since cats are obligate carnivores and dogs are not.

It would be helpful to see the starting position in Figure 1.

Success should also be defined in the list of coded behaviors. Exact Kappas should be reported for each coded behavior that is analyzed.

In Figure 5, I am confused by “followed T1 at least once.” Wouldn’t this also include those subjects that followed T1 in both subsequent trials?

Please indicate in which groups the dogs looked back at the owners, as you did for cats. Perhaps the frequencies of looking back across conditions could be compared per species.

The writing is a bit awkward, likely due to the authors not being native English writers. Just a few of the issues are listed below. I commend the authors for writing in a foreign language but they might want to have a native writer proof-read the revised paper.

Line 64, there is an extra space after [15].

Line 65, should be “have been testing..”

Line 73-74 awkward and needs to be rephrased.

Line 77, “a priori own experience” is also an awkward phrasing

Author Response

REVIEWER 1

It is great that the authors compared the performance of cats to dogs directly in the same task. I like that they noted whether both species looked to the human and considered multiple interpretations of this finding.

This is a nice sample size for both cats and dogs.

RESPONSE: Thank you for these comments, we truly appreciate it! Direct comparison of cats and dogs was one of our main goals with this study. Testing considerable Ns of especially the cats was rather a grueling task, as these had to visited at their owners’ homes one by one.

The demonstration manipulation should be justified and described in the introduction. The use of a laser pointer instead of a conspecific demonstration should be more clearly justified in the introduction as well. The authors should address how their procedure deviates from more traditional versions of the detour task using both transparent and opaque barriers.

RESPONSE: Thank you for this useful comment. Very briefly, we opted for NOT using live (conspecific or human) demonstrators in this study because it would be a totally different study, as there is a still open question whether cats would show such strong reliance on social learning as dogs do. This is definitely an intriguing topic for a follow up study! In the present work we wanted to collect more basic knowledge about (1) cats’ performance in the classic detour task, which has been already tried with dogs several times. Additionally, (2) we wanted to use with both species an asocial ‘demonstration’ of detour, where we opted for the laser dot. The original idea behind this was that as both cats and dogs like to chase laser light, this stimulus might have a different effect on their detour performance, as cats and dogs show different hunting strategies. In our opinion, these thoughts on the usage of laser light stimulus are clearly emphasized in lines 213-219.

Regarding our choice to use the V-shaped transparent obstacle for the detour task, this was motivated by our plan to provide a ‘hard enough’ task for the subjects (according to the literature, detours around opaque barriers seem to be easier). Additionally, the direct predecessor of this method (published in several papers from Pongrácz and colleagues), also used a similar device on dogs (although twice as long, due to the outdoor conditions in those studies).

We added parts of the above mentioned answers to the Introduction as well (lines 193-201).

The authors should discuss whether and why cats and dogs may differ in terms of win stay win shift foraging strategies. Perhaps this makes sense since cats are obligate carnivores and dogs are not.

RESPONSE: Thank you for the very interesting comment. We found the inclusion of win-stay strategy as a potential explanation to our findings as an excellent one. In our opinion, dogs are typically following the win-stay/lose-shift strategy, as they are rather scavangers than real predators anymore (supported by Byrne, M., Bray, E., Maclean, E., & Johnston, A. (2020). Evidence for win-stay-lose-shift in puppies and adult dogs. In CogSci.). In dogs, our results support this notion (they tend to stay on the side of the fence where previously they succeeded). Cats on the other hand, typical ambush predators, hunting versatile and unpredictable prey, and thus for them, sticking to an already ‘exploited’ spot would not be the best strategy. The essence of this response is incorporated now to the Discussion (lines 804-807 and 830-832).

It would be helpful to see the starting position in Figure 1.

RESPONSE: We added another photo, where the starting position is clearly visible.

Success should also be defined in the list of coded behaviors. Exact Kappas should be reported for each coded behavior that is analyzed.

RESPONSE: Thank you for requesting these details. ‘Success’ was added to the list of behaviors in Table 1. We added the Kappa values for cats’ and dogs’ detour latencies (lines 605-606), the frequency of following the detour direction in T1 (cats and dogs, lines 653-654).

In Figure 5, I am confused by “followed T1 at least once.” Wouldn’t this also include those subjects that followed T1 in both subsequent trials?

RESPONSE: Thank you for catching this. You are absolutely right. The figure legend was amended, so it now reads “Followed T1 side once”.

Please indicate in which groups the dogs looked back at the owners, as you did for cats. Perhaps the frequencies of looking back across conditions could be compared per species.

RESPONSE: We added the following details to the results (lines 672-674) “Out of the 38 dog participants, 13 animals looked back at the owner, and four dogs did it more than once (control 5, straight pointing group 5, detour pointing group 3).”

The writing is a bit awkward, likely due to the authors not being native English writers. Just a few of the issues are listed below. I commend the authors for writing in a foreign language but they might want to have a native writer proof-read the revised paper.

RESPONSE: Thank you for this comment. Yes, it is rather hard to communicate on other than our mother tongue, especially when both the scientific content and the readers’ experience should theoretically be impeccable. We asked a native English speaker colleague now for proof reading the text. Hopefully there is some improvement – at least there is a lot of corrections J

Line 64, there is an extra space after [15].

Line 65, should be “have been testing..”

Line 73-74 awkward and needs to be rephrased.

Line 77, “a priori own experience” is also an awkward phrasing

RESPONSE: Thank you for the suggestions, these were all fixed.

Reviewer 2 Report

The authors found interesting species differences in the problem solving behavior of companion cats and similarly sized dogs in the detour task. The study is unique in that few studies have directly compared the cognitive skills of these two species. My specific comments are listed below.

Simple Summary:

Line 15: Change “form” to “from.”

1. Introduction:

The Introduction provides a comprehensive review of the literature. I like that the authors put the detour task in an ecological context and include their predictions for study outcomes. 

2. Materials and Methods:

Did you ask cat owners if they had previously used laser pointers when playing with their cats? Could previous exposure to a laser pointer affect you results?

I fully understand why you tested cats in their homes, but wonder why you did not test dogs in their homes as well (even given the evidence that dogs can be reliably tested in unfamiliar conditions as long as the owner is present). Was it also because you wanted to be able to compare your present findings to your previous work with dogs (and findings of other researchers), which was conducted under laboratory conditions? It would be good to make a stronger case than that made in lines 184-189.

The description of study procedures is very clear and sufficiently detailed to allow replication.

Table 1, Looking back (frequency), Description: should “did not reach” be changed to “reached”?

Results: 

Figures should be stand alone, able to be understood by someone without reading the text. As such, please add units “(seconds)” after the label “Time” for the X-axis in Figures 2-4.

Line 355: Does “static”  mean the straight pointing group when the laser was pointing at the target for 3 seconds?

Discussion:

Line 359: I don’t think you can say “We tested companion cats and dogs among identical indoor conditions…” given that cats were tested in their homes and dogs in the laboratory. Instead, maybe state something like the following:  “We tested companion cats and dogs in similar indoor conditions, using identical procedures in the classic detour task around a V-shaped transparent wire-mesh fence.” I also would like to see the use of different testing environments (home versus laboratory) listed as a potential study limitation in lines 477-483.

I enjoyed reading this paper and hope the authors find my comments helpful.

Author Response

REVIEWER 2

The authors found interesting species differences in the problem solving behavior of companion cats and similarly sized dogs in the detour task. The study is unique in that few studies have directly compared the cognitive skills of these two species. My specific comments are listed below.

RESPONSE: Thank you very much for the nice compliment. We are happy that our message went through.

Simple Summary:

Line 15: Change “form” to “from.”

RESPONSE: Fixed.

  1. Introduction:

The Introduction provides a comprehensive review of the literature. I like that the authors put the detour task in an ecological context and include their predictions for study outcomes.

RESPONSE: Awesome! This was exactly our goal! Thank you.

  1. Materials and Methods:

Did you ask cat owners if they had previously used laser pointers when playing with their cats? Could previous exposure to a laser pointer affect you results?

RESPONSE: Yes, we had a short questionnaire where we collected the basic details of the subjects. In this we also asked whether the owner ever tried playing with the subject by using a laser pointer. Most owners asked ‘yes’. However, the results showed that laser pointer did not have an effect on the detour performance. By our opinion, the moving laser could have an effect but only if the subjects could follow it while it was moving around the fence. However, that would been a totally different task.

I fully understand why you tested cats in their homes, but wonder why you did not test dogs in their homes as well (even given the evidence that dogs can be reliably tested in unfamiliar conditions as long as the owner is present). Was it also because you wanted to be able to compare your present findings to your previous work with dogs (and findings of other researchers), which was conducted under laboratory conditions? It would be good to make a stronger case than that made in lines 184-189.

RESPONSE: Thank you for the question. The most relevant reason for NOT testing the dogs at their owners’ homes was that since well-socialized dogs behave naturally anywhere they go with their owners, researchers simply opt for the most convenient method: testing dogs at the laboratory, or at dog schools, etc. Unlike cats, where most individuals strongly stress when being taken away from home, dogs are easily accommodated with novel locations in the presence of their owners. One more issue might emerge when researchers want to test dogs at their homes: some dogs behave territorially at home, which can cause problems. We added this detail to the Methods (Lines 301-304). There are also thoughts on this topic in the Discussion (Lines 892-911).

The description of study procedures is very clear and sufficiently detailed to allow replication.

RESPONSE: Thank you, we are glad that our efforts paid off.

Table 1, Looking back (frequency), Description: should “did not reach” be changed to “reached”?

RESPONSE: Thank you for spotting this – we corrected it.

Results:

Figures should be stand alone, able to be understood by someone without reading the text. As such, please add units “(seconds)” after the label “Time” for the X-axis in Figures 2-4.

RESPONSE: The figures were amended. Thank you for the suggestion.

Line 355: Does “static”  mean the straight pointing group when the laser was pointing at the target for 3 seconds?

RESPONSE: Thank you for noticing the bad terminology – we changed “static” to “straight”, which is the correct name of the experimental treatment here.

Discussion:

Line 359: I don’t think you can say “We tested companion cats and dogs among identical indoor conditions…” given that cats were tested in their homes and dogs in the laboratory. Instead, maybe state something like the following:  “We tested companion cats and dogs in similar indoor conditions, using identical procedures in the classic detour task around a V-shaped transparent wire-mesh fence.” I also would like to see the use of different testing environments (home versus laboratory) listed as a potential study limitation in lines 477-483.

RESPONSE: Thank you for the suggested sentence, we applied it as you have suggested (both in the Abstract and in the Discussion). Now we also mention the difference between the testing locations as a potential limiting factor (lines 883-886). However, as the results showed that dogs actually performed rather well in the experiments, we are not sure what would be the effect of testing them in the lab, compared to a theoretical test at their homes.

I enjoyed reading this paper and hope the authors find my comments helpful.

RESPONSE: We are glad that you showed such a supportive attitude towards our work. Thank you very much!

Reviewer 3 Report

This manuscript describes a fascinating experiment comparing domestic cats' and dogs' cognitive ability to solve a detour task. Overall, cats' and dogs' performance was similar. But dogs were faster than cats on Trial 1 and gained speed up to Trial 3. Moreover, from trial to trial, dogs used the same successful pathway. Cats did not.

The literature review is excellent. Similarly, the methods are well-described and easy to replicate. The statistical analyses are flawless, and the authors nicely and precisely present the results. 

I have only a few points to highlight, which the authors might like to consider for their final version:

1- I was surprised that verbal encouragement was allowed during each trial. This procedure can explain why some dogs (and a few cats) looked back at their owner during the detour. Consequently, I would suggest removing all data about "looking back to the owner." Again, I really believe this procedure interfered with the dogs' and cats' behaviour.

2- In the laser condition, I suspect cats did not use it because it was used as a demonstration (to show the path from the cat's initial position to the food location). The cats were not allowed to chase the laser. If they would, they probably would have followed it. I suggest adding this possibility to the general discussion.

3- I am a bit more concerned about the interpretation suggesting that cats might be better (or more flexible) than cats in solving the detour task. This interpretation is mostly based on the observation that dogs tended to follow their previous successful path (cats did not). Someone could argue that dogs demonstrated a better learning capacity than cats because they could remember and use their last successful action. I suggest adding this possibility to the general discussion too.

Author Response

REVIEWER 3

This manuscript describes a fascinating experiment comparing domestic cats' and dogs' cognitive ability to solve a detour task. Overall, cats' and dogs' performance was similar. But dogs were faster than cats on Trial 1 and gained speed up to Trial 3. Moreover, from trial to trial, dogs used the same successful pathway. Cats did not.

The literature review is excellent. Similarly, the methods are well-described and easy to replicate. The statistical analyses are flawless, and the authors nicely and precisely present the results. 

RESPONSE: Thank you so much for this encouraging comment. It means a lot to us!

I have only a few points to highlight, which the authors might like to consider for their final version:

1- I was surprised that verbal encouragement was allowed during each trial. This procedure can explain why some dogs (and a few cats) looked back at their owner during the detour. Consequently, I would suggest removing all data about "looking back to the owner." Again, I really believe this procedure interfered with the dogs' and cats' behaviour.

RESPONSE: Thank you for this interesting comment. Verbal encouraging and calling of the subjects’ attention was an inherent element in all the problem solving experiments previously conducted by this team in the past 20-some years in case of not only dogs (e.g., in case of detour task, see Pongrácz et al., 2001, 2003, 2004), but cats also (Pongrácz et al., 2019; Pongrácz & Onofer, 2020). Importantly, it was shown that companion dogs especially, do not perform well without their owners’ encouragement (Topál et al., 1997). Therefore, especially in case of dogs, not allowing verbal encouragement from the owners would create an unnatural situation, which could potentially lead to artificially low performance.

We should also mention that dogs’ ‘looking back’ to the humans in case of the detour task was found to be connected to the difficulty of the task (among others, Pongrácz et al., 2001). In other words, dogs look back at the owner if they have problems with detouring – and not because the owner uses verbal encouragement.

Cats’ responsivity to human verbal utterances was found by Japanese researchers several occasions (e.g., Saito & Shinozuka, 2013; Saito et al., 2019).

However, we think that the Reviewer’s comment about the potentially different response pattern to verbal encouragement in cats and dogs is a valid detail to be placed among the limitations of the study (Lines 857-867).

2- In the laser condition, I suspect cats did not use it because it was used as a demonstration (to show the path from the cat's initial position to the food location). The cats were not allowed to chase the laser. If they would, they probably would have followed it. I suggest adding this possibility to the general discussion.

RESPONSE: Absolutely correct! This also what we thought as a potential explanation. Of course, this became clear only after the tests were run and analyzed. This potential explanation is now mentioned in the Discussion (Lines 883-884). Thank you.

3- I am a bit more concerned about the interpretation suggesting that cats might be better (or more flexible) than dogs in solving the detour task. This interpretation is mostly based on the observation that dogs tended to follow their previous successful path (cats did not). Someone could argue that dogs demonstrated a better learning capacity than cats because they could remember and use their last successful action. I suggest adding this possibility to the general discussion too.

RESPONSE: Thank you for this alternative explanation, definitely worth to include it to the manuscript. It is an interesting question, whether following the previously chosen path would be the proof of a better learning performance of the subject, or perhaps this is a strategy that is chosen when the subject had more difficulty with the detour problem. In our earlier research (Pongrácz, P., Miklósi, Á., Kubinyi, E., Topál, J., & Csányi, V. (2003). Interaction between individual experience and social learning in dogs. Animal Behaviour65(3), 595-603.) we found that, for example, dogs only learned to follow the (human) demonstrator’s path during the detour when they did not have own successful attempts previously. Otherwise they opted for their own previous detour direction.

We tend to agree with the Reviewer 3 that the dogs’ improvement in their detour latency and their tendency to follow the previously prevailing path might have a connection between.

We added this thought to the Discussion (lines 835-839).